# Intracellular pH Regulation of Skeletal Muscle in the Milieu of Insulin Signaling

**DOI:** 10.3390/nu12102910

**Published:** 2020-09-23

**Authors:** Dheeraj Kumar Posa, Shahid P. Baba

**Affiliations:** 1Diabetes and Obesity Center, University of Louisville, Louisville, KY 40202, USA; spbaba01@louisville.edu; 2Christina Lee Brown Envirome Institute, University of Louisville, Louisville, KY 40202, USA

**Keywords:** carnosine, chronic kidney disease, diabetes, glycolysis, histidyl dipeptides, insulin signaling, intracellular pH, obesity

## Abstract

Type 2 diabetes (T2D), along with obesity, is one of the leading health problems in the world which causes other systemic diseases, such as cardiovascular diseases and kidney failure. Impairments in glycemic control and insulin resistance plays a pivotal role in the development of diabetes and its complications. Since skeletal muscle constitutes a significant tissue mass of the body, insulin resistance within the muscle is considered to initiate the onset of diet-induced metabolic syndrome. Insulin resistance is associated with impaired glucose uptake, resulting from defective post-receptor insulin responses, decreased glucose transport, impaired glucose phosphorylation, oxidation and glycogen synthesis in the muscle. Although defects in the insulin signaling pathway have been widely studied, the effects of cellular mechanisms activated during metabolic syndrome that cross-talk with insulin responses are not fully elucidated. Numerous reports suggest that pathways such as inflammation, lipid peroxidation products, acidosis and autophagy could cross-talk with insulin-signaling pathway and contribute to diminished insulin responses. Here, we review and discuss the literature about the defects in glycolytic pathway, shift in glucose utilization toward anaerobic glycolysis and change in intracellular pH [pH]_i_ within the skeletal muscle and their contribution towards insulin resistance. We will discuss whether the derangements in pathways, which maintain [pH]_i_ within the skeletal muscle, such as transporters (monocarboxylate transporters 1 and 4) and depletion of intracellular buffers, such as histidyl dipeptides, could lead to decrease in [pH]_i_ and the onset of insulin resistance. Further we will discuss, whether the changes in [pH]_i_ within the skeletal muscle of patients with T2D, could enhance the formation of protein aggregates and activate autophagy. Understanding the mechanisms by which changes in the glycolytic pathway and [pH]_i_ within the muscle, contribute to insulin resistance might help explain the onset of obesity-linked metabolic syndrome. Finally, we will conclude whether correcting the pathways which maintain [pH]_i_ within the skeletal muscle could, in turn, be effective to maintain or restore insulin responses during metabolic syndrome.

## 1. Introduction

Diabetes and obesity-related diseases afflict 425 million people worldwide and have emerged as a significant cause of mortality and morbidity in both the developed and developing countries. Type 2 diabetes (T2D) is associated with decrease in number and defects in the function of insulin-producing pancreatic β-cells. However, the decreased response of skeletal muscle to insulin is evident even before pancreatic β-cell failure [1,2]. Skeletal muscle is the principal site for insulin-stimulated glucose uptake, which consumes approximately 70–80% of the glucose via insulin-dependent mechanisms [3]. Hence, defects in insulin-stimulated muscle glucose uptake are considered a principal component of typical obesity-associated insulin resistance [4,5]. Glucose homeostasis in the muscle is a complex interplay between insulin signaling and glucose utilization. However, a consistent finding is that maintaining proper insulin responses of the IRS1–PI3K–AKT signaling pathway within skeletal muscle is central to maintaining glucose homeostasis. This pathway is markedly impaired in the muscle of humans with type 2 diabetes (T2D), healthy glucose-tolerant offspring of parents with T2D, obese non-diabetics, people with non-alcoholic fatty liver disease and chronic kidney disease (CKD) [6,7,8,9,10,11,12]. Defects in insulin-signaling pathway are traceable to defective insulin receptor (IR) kinase activity, a shift from tyrosine to serine phosphorylation in IRS1 and blunted IRS1-associated P13K activity, followed by a subsequent decrease in AKT phosphorylation [7]. Much attention in the field has been focused on the defects of insulin-mediated metabolic actions via the IRS1–PI3K–AKT signaling pathway in the muscle. However, in addition to defects in this signaling pathway, numerous reports suggest that other pathways, such as inflammation [13,14,15,16], lipid peroxidation products [17], oxidative stress [18,19,20,21,22,23], a decrease in intracellular pH [pH]_i_ [9,24] and autophagy [25,26], could cross-talk with insulin-signaling pathway and either cause or exacerbate insulin resistance in the muscle. In this article, we will highlight a few issues, which are central to the gap in knowledge about whether the defects in glycolysis, within the muscle are causal or secondary to insulin resistance. We will discuss, whether the increase in interconversion of pyruvate to lactic acid, which subsequently dissociates to lactate and hydrogen (H^+^) ions, in the skeletal muscle of T2D humans, could contribute to a decrease in [pH]_i_. We discuss, whether the defects in H^+^ ion transporters and depletion of intracellular histidyl dipeptides, in the skeletal muscle could contribute towards decrease in [pH]_i_ and be a causative factor of insulin resistance. Furthermore, we will highlight studies suggesting the effect of a decrease in [pH]_i_ on the intermediates of glycolytic pathway and protein degradation. Finally, we will highlight, whether correcting defects in pathways which maintain [pH]_i_ within the muscle, could alleviate insulin resistance and improve insulin responses during metabolic syndrome.

## 2. Defects in Glycolytic Pathway During Type 2 Diabetes

Complete oxidation of glucose under aerobic conditions, results in the net production of 38 molecules of ATP. The first step after the entry of glucose into a cell is rapid phosphorylation to glucose-6-phosphate by hexokinases, which entraps glucose. In mammalian tissues, there are at least four isoforms of hexokinases (I–IV). Hexokinase I (HKI) is relatively ubiquitous, but hexokinase II (HKII) is abundantly expressed in the skeletal muscle and contributes to maximum HK activity [27]. Dynamic positron emission tomography (PET) imaging of the skeletal muscle in obese humans, with type 2 diabetes (T2D) showed that insulin resistance strongly correlates with impairments in glucose transport and the efficiency of glucose phosphorylation [28]. Consistent with these observations, *HKII* gene expression is particularly low in the skeletal muscle of patients with T2D [29] and the response of *HKII* to insulin is blunted in the skeletal muscle of T2D patients [30]. Exercise increases mRNA levels of *HKII* in the skeletal muscle of obese and diabetic humans; however, HKII enzyme activity in response to exercise remains largely unaffected [31]. Even though HKII is the first rate-limiting enzyme in glycolytic pathway and glucose phosphorylation is diminished in the muscle of humans with T2D [28], studies with mice heterozygous for targeted disruption of *HKII* gene and fed with high-fat diet, were neither insulin-resistant nor glucose-intolerant [32]. However, studies with mice that overexpress *HKII* in the skeletal muscle increases glucose-6-phosphate levels, alleviates some markers of metabolic syndrome, such as normalizes fasting blood glucose and improves exercise-stimulated glucose uptake in the tissues of high fat-fed mice [27,33].

In addition to HK, phosphofructokinase-1 (PFKM) is the second rate-limiting enzyme of glycolysis, which converts fructose-6-phosphate to fructose-1,6-bisphosphate [34]. There are at least three different isoenzymes found in the liver, skeletal muscle and platelets [34]. The inherent deficiency of the muscle *PFKM* gene leads to autosomal, recessively inherited glycogen storage VII, Tarui’s disease, which causes hemolysis, excess glycogen storage, exercise-induced myopathy and altered insulin action [35]. In a study of a core family of four members, where the father and older son were homozygous and the mother and younger son had a heterozygous *PFKM* gene deficiency, both parents and the older son showed diminished glucose disappearance and insulin resistance, and the younger son had moderate insulin resistance. These studies suggested that the deficiency of *PFKM*, or a decrease in PFKM activity, could be the metabolic sequelae to predispose for T2D [36]. Paradoxically, expression quantitative trait loci studies (eQTLs) performed with the skeletal muscle of T2D subjects, found that *PFKM* gene expression was increased and associated with reduced glucose uptake and insulin resistance [37]. Although *PFKM* expression is increased within the muscle of T2D patients, *in vitro* activity assays suggests that PFKM enzyme activity is susceptible to changes in pH, within a range of pH 6.9–7.1. PFKM enzyme activity decreases significantly at a lower pH [38]. Based on reports showing that buffering capacity within the muscle of humans with T2D is decreased compared with healthy subjects [39], it is suggested that a decrease in [pH]_i_ could decrease PFKM enzyme activity in the skeletal muscle of T2D patients.

The final rate-limiting step of glycolysis is catalyzed by pyruvate kinase, which yields pyruvate that crosses the mitochondrial membrane to enter the Krebs cycle. Pyruvate dehydrogenase complex (PDC) is a significant determinant for the fate of pyruvate, which converts pyruvate to acetyl-CoA. PDC is deactivated by pyruvate dehydrogenase kinase (PDK) and ATP, when the cellular energy stores are high [40]. Although PDC activity is decreased in the skeletal muscle of humans with T2D [41], studies with mice show that constitutive PDC overexpression in the skeletal muscle causes insulin resistance. The decrease in insulin sensitivity by PDC activation was partly mediated by increase in glucose oxidation, reduction of fatty acid utilization and accumulation of intramyocellular lipids [42]. Because the decrease in PDC activity could steer the pyruvate utilization from mitochondria toward lactate formation, indeed in patients with T2D, there is a fourfold increase in both lactate and pyruvate interconversion within skeletal muscle [43]. Extensive epidemiological studies from humans show that lactate levels are increased in the T2D subjects. Lactate levels monitored in the blood of T2D patients over 24 h periods were significantly higher, when compared with healthy subjects [44]. Similarly, a community-based prospective cohort of approximately 16,000 participants showed that the upper quartile of baseline lactate in serum directly correlates with increased risk of T2D [45,46]. Since skeletal muscle utilizes 60–70% of the total glucose and muscle contributes to 25% of lactate production [47,48], increased interconversion of pyruvate to lactate formation in the muscle might contribute to higher lactate levels observed in the blood of T2D patients.

Lactate is formed from the dissociation of lactic acid to lactate and H^+^ ions. An increase in H^+^ ion production during T2D could decrease [pH]_i_ within the muscle. In patients with T2D, compared with normal humans, the decrease in [pH]_i_ within the skeletal muscle is significantly faster during exercise, suggesting a lower buffering capacity and overwhelming of the processes which regulate [pH]_i_ [39]. Alternatively, lactate also contributes to decrease in insulin-induced glucose uptake. Studies with rats showed that lactate infusion suppresses glycolysis and glucose uptake in the skeletal muscle. Lactate diminishes the ability of insulin to stimulate IRS-1 or IRS-2-associated PI3K activities and AKT/protein kinase B activity, without any change in GLUT4 protein content [49].

Nevertheless, these data indicate that there are derangements within the key regulatory steps of glycolytic pathway in the muscle during T2D. However, studies with mice overexpressing HKII and PDC enzymes, were unable to rescue the insulin resistance and obese metabolic phenotype. Taken together these studies suggest that either the decrease in glycolytic enzyme activity is a secondary effect of insulin resistance, or overexpression of these enzymes in skeletal muscle activates compensatory pathways, which cause insulin resistance. Although heterozygous *PFKM* deficiency leads to insulin resistance, whether the decrease in buffering capacity within the muscle of humans with T2D could decrease PFKM enzyme activity and lead to onset of insulin resistance needs to be studied.

## 3. Acidosis and Insulin Resistance

Numerous in vitro and in vivo studies implicate the involvement of acid–base imbalance in the worsening of insulin’s glucose-lowering effects [50,51,52]. Urine acidification is a common feature seen in insulin-resistant individuals. Humans with metabolic syndrome who were free from kidney stones had a lower urine pH compared with participants without metabolic syndrome and the mean 24 h pH decreased from 6.15 to 5.69 [53,54]. In healthy humans, the slightest degree of metabolic acidosis triggers insulin resistance. Hyperglycemic and euglycemic studies in normal humans indicated that NH_4_Cl injections impairs glucose metabolism and tissue sensitivity to insulin [55]. Excessive ketone body production in diabetic patients is considered a significant source of acidosis and insulin resistance in muscle [56,57]. A large epidemiological National Health and Nutrition Examination Survey (NHANES) showed that lower bicarbonate plasmatic levels and higher anions are independently associated with decreased insulin sensitivity [58]. Metabolic acidosis is also a common complication in patients with end-stage chronic renal failure, which induces insulin insensitivity and glucose intolerance by impairing PI3K activity in the muscle [59,60]. Studies from human patients on hemodialysis with insulin resistance and patients with CKD, showed that alkalizing treatment with oral sodium bicarbonate significantly improved insulin sensitivity and secretion [61,62].

Acid-induced insulin resistance has been reported in several cell types, particularly in muscle myoblasts [24,63]. Hayata et al. reported that in L6 myoblasts, insulin binding to insulin receptors was downregulated approximately 50% at pH 6.8, suggesting that acidosis has the potential to diminish glucose entry, the first step of insulin-signaling pathway. By using radiolabeled insulin, the authors showed that a decrease in insulin binding to the IR receptor affects the phosphorylation of Tyr1146, Ser473 and Thr308 of IR [63]. Although these studies suggest that the decrease in insulin responses was dependent on decrease in extracellular pH, whether there was a reduction in [pH]_i_ was not studied. To examine whether intracellular acidification could affect insulin response, Franch et al. compared the effects of long- and short-term acidic conditions (pH 7.1) on muscle myoblasts. Incubation of myoblasts for shorter periods (10 min) did not affect insulin responses, whereas myoblasts incubated under acidic conditions for 24 h had a marked reduction in PI3K activity and acidification. A decrease in PI3K activity caused a marked reduction of AKT phosphorylation and the upstream PI3K signaling process, whereas IRS abundance and phosphorylation remained unchanged [24].

The maintenance of [pH]_i_ is essential for cellular functions. Even the slightest deviation from the acceptable range of pH (from 7.38 to 7.42) alters the tertiary structure of proteins, causes protein denaturation and reduces the catalytic activity of enzymes. For example, during CKD, acidification of the skeletal muscle stimulates the formation of ubiquitinated proteins [59]. Recent reports suggest that the decrease in [pH]_i_ activates autophagy, which can be suppressed by an increase in [pH]_i_ [64]. Autophagy is a key regulatory intracellular degradation process, which maintains cellular homeostasis by removing protein aggregates and damaged organelles during development and nutrient stress [65]. In humans with T2D, gene expression profiles of autophagy-related genes, such as *ATG14* and *LCBII*, are decreased in the skeletal muscle, indicating that defective autophagy could contribute to diminished insulin responses [66]. Both the activation and deficiency of autophagic flux in the skeletal muscle has been implicated in the pathogenesis of insulin resistance. Genetic deletion of *Atg7* (autophagy-related gene 7) in the mice’s skeletal muscle decreased fat mass and protected from diet-induced obesity by promoting the release of myokine Fgf21 [26]. On the contrary, studies with mice containing knock-in mutations of BCL2 phosphorylation sites in the skeletal muscle, which decreases exercise induced autophagy, altered glucose metabolism during exercise and impaired exercise-mediated protection against high-fat diet-induced metabolic syndrome [25]. Similarly, studies with mice, where autophagy was activated by exercise [67] and treatments with uncylated ghrelin [68] and adiponectin [69] restored insulin signaling.

Taken together, there is extensive evidence that the decrease in both extra and intracellular pH could diminish insulin responses. However, the mechanisms by which the drop-in [pH]_i_ affects insulin signaling responses and the contribution of decrease in [pH]_i_ towards activation or derangements in autophagic flux is not clear.

## 4. Maintenance of Intracellular pH ([pH]_i_) in the Muscle

Regulation of [pH]_i_ within the muscle is a complex process, which is maintained by several membrane transporters and endogenous buffers [70]. In the muscle, pH homeostasis is maintained by a complex transporter system that involves the Na^+^/H^+^ exchanger (NHE), Na^+^-dependent and independent bicarbonate systems (NBC) and monocarboxylate transporters (MCT1 and 4) [70,71,72,73,74]. In the skeletal muscle, the NHE system is considered the most important regulatory system, which is active at resting levels. However, a minor fraction of H^+^ ions are released by the NHE during exercise [48]. In rats fed with low doses of streptozotocin and a high-fat diet, NHE expression was decreased in the soleus and extensor digitorum longus muscles, which was increased by endurance training [75]. There are at least two isoforms of the NBC transport systems, NBCe1 and NBCe2, present in the human skeletal muscle; however, the contribution of these transporters to pH regulation is not clear [76]. Studies with diabetic rats showed that NBC expression remains unchanged and endurance training increases NBC expression in the muscle [75].

In addition to the NHE and NBC, two monocarboxylate transporters, MCT1 and MCT4, present in the muscle predominantly facilitate H^+^ ions and lactate efflux (symporter) across the plasma membrane during exercise [72]. MCT1 is ubiquitously expressed, whereas MCT4 is mainly expressed in the glycolytic muscle fibers. Skeletal muscle is made up of different bundles of muscle fibers, which are broadly classified as slow-twitch (Type 1) and fast-twitch (Type 2) fibers. Based on the myosin heavy chain gene expression (*MYH*), fast-twitch is further classified as Type 2a, 2X and 2B. The skeletal muscle fibers vary in energy production, with Type 1 and 2A primarily using the oxidative metabolism and type 2X and 2B fibers using the glycolytic metabolism [77,78]. Numerous studies show that MCT1 and MCT4 protein expression is increased in the skeletal muscle after exercise training and electrical stimulation in healthy humans [79,80,81]. MCT1 expression is increased after endurance training and its expression is correlated with the oxidative capacity [82], whereas MCT4 expression is increased following intensive exercise and related to the indexes of glycolytic metabolism [83]. Because MCT1 is mostly found in oxidative fibers and displays a higher affinity for L-lactate compared with MCT4, it has been suggested that MCT1 mainly handles the uptake of lactate and H^+^ ions, whereas MCT4 is involved in lactate and H^+^ efflux [84,85,86,87]. However, all the MCTs can operate in both directions, involving the efflux and influx of lactate and H^+^ ions. In humans with T2D, MCT1 expression is decreased in the skeletal muscle compared with healthy humans [88]. Studies with heterozygous MCT1^+/−^ mice show that the [pH]_i_ within the muscle at rest was higher compared with the wild type mice, but the drop in [pH]_i_ was higher in the MCT1^+/−^ mice during the initial minutes of exercise. Hence, MCTI is involved in the homeostatic control of pH within the skeletal muscle both at rest and during exercise [71]. MCT1^+/−^ mice displayed normal insulin sensitivity, whereas MCT1^+/−^ mice fed with a high-fat diet were resistant to diet-induced obesity, insulin resistance and glucose intolerance. This phenotype was associated with reduced food intake and decreased intestinal absorption under high-fat diet conditions [89]. Given that MCT1 handles H^+^ influx [84,85,86,87], it could be speculated that MCT1 deletion, diminishes the uptake of H^+^ ions into the muscle, maintains alkaline pH and thus preserves the insulin responses during high-fat feeding. In addition to these transport systems, the skeletal muscle contains at least four isozymes of carbonic anhydrase (CA II, III, IV and V) that accelerate the removal of acid as CO_2_ [90]. Proteomic profiling of crude muscle extracts from the non-obese Goto–Kakizaki rat model of T2D identified that the expression of CA III had the highest decrease compared with the non-obese mice [91]. Since the CA III isoform is one of the faster enzymes which helps regulate [pH]_i_, the decrease in its expression could also decrease the removal of acid in diabetic muscle. Indeed studies with CA III knock out mice showed that the drop in [pH]_i_ following intense stimulation of the gastrocnemius muscle was significantly more in the CA III knock out mice compared with the wild type mice [92]. 

Taken together, these studies suggest that the essential components of pH regulatory system are imbalanced in T2D muscle. Further the improvement in insulin action by endurance training could be facilitated by increased expression of both NHE and MCT1 transporters in the skeletal muscle [93,94,95,96,97], which could enhance the removal of H^+^ ions and thus improve buffering capacity.

## 5. Histidyl Dipeptides and Intracellular pH [pH]_i_ Regulation

Histidyl dipeptides have a pKa value close to the physiological pH (6.8–7.1), compared with the bicarbonate (pKa 6.3), inorganic phosphate (pKa 7.2) and histidine (pKa 6.2) [98], which renders them ideally suited to act as intracellular buffers within the pH transit range of skeletal muscle. Theoretical estimates predict that these dipeptides could contribute to approximately 7–40% of muscle buffering capacity [99,100]. In comparison with high molecular weight transporters and enzymes, which possess lower intracellular mobility and restrict their capacity to buffer protons and correct the [pH]_i_, histidyl dipeptides are small molecular weight dipeptides with a molecular weight range of 227–241 Da. These dipeptides diffuse reversibly two or more orders of magnitude faster than proteins across the cell [101]. Skeletal muscle is the largest reservoir of histidyl dipeptides, and approximately 10–20 mM levels of these dipeptides are present in the muscle [101,102]. Among the naturally occurring histidyl dipeptides, carnosine is the most common dipeptide present in human skeletal muscle, whereas its methylated analogues, anserine (β-alanine-N^π^-histidine) and balenine (β-alanine-N^τ^-histidine) are largely found in other mammalian and avian species [101,102,103].

Carnosine is synthesized via the ATP grasp protein carnosine synthase (Carns1), which ligates the non-proteinogenic amino acid β-alanine with histidine to form carnosine and further methylates to anserine by carnosine methyltransferase [104,105,106,107]. In addition to [pH]_i_ buffering [101,108], these dipeptides also scavenge lipid peroxidation products, such as 4-hydroxy trans-2-nonenal (HNE) [109,110], quench reactive oxygen species (ROS) such as singlet oxygen [111] and chelate first transition metals [112]. Recent reports from our laboratory and others showed that carnosine levels are increased upon exercise and significantly depleted in the muscle of normal humans and T2D patients, respectively [102,113]. We also tested whether the levels of histidyl dipeptides are affected in the skeletal muscle by high-fat feeding in rodent models. For these studies, we performed a pilot study and fed the wild type C57 mice (7–8 weeks) with a high-fat and high-sucrose (HFHS) diet for 12–14 weeks. Gastrocnemius muscle from the normal chow (NC)- and HFHS-fed mice were analyzed by LC–MS for carnosine and anserine levels. In parallel with the observations in T2D humans that histidyl dipeptides are depleted in the gastrocnemius muscle [113], we found that carnosine levels in the gastrocnemius muscles of HFHS-fed mice were significantly decreased compared with the NC-fed mice (Figure 1). Taken together, these reports showing that carnosine levels change under divergent conditions suggest that histidyl dipeptides may be essential for regulating specific metabolic processes such as [pH]_i_ under physiological and pathological conditions.

It has been reported that carnosine supplementation in different animal models of metabolic syndrome could alleviate some markers of metabolic syndrome. Studies with high fat-fed mice supplemented with histidine, β-alanine or carnosine showed that histidine and carnosine supplementation reduced the activities of lipogenic enzymes and sterol regulatory element-binding proteins in the liver. These changes were also accompanied with improved insulin sensitivity and hyperinsulinemia [114]. Carnosine feeding to obese Zucker rats enhanced the extrusion of lipid peroxidation products, such as 4HNE, by forming carnosine aldehyde conjugates, diminishing carbonyl stress, dyslipidemia and hypertension [109]. Similarly, carnosine supplementation to *db/db* mice increased the pancreatic islets sizes and insulin release [115]. In parallel with the reports from animal models of diabetes, recent studies with obese insulin-resistant humans showed that carnosine supplementation modestly attenuates insulin release and enhances the extrusion of reactive aldehydes in urine [116,117]. However, a major obstacle for clinical applicability with these dipeptides is that they are hydrolyzed by carnosinase (CNDP1) present in the serum, which diminishes their bioavailability at the site of injury [118]. To circumvent this challenge, several groups tested the carnosine analogues, which are resistant to hydrolysis, such as D-carnosine. However, these carnosinase-resistant enantiomers have poor absorption and failed in testing [119]. Similarly, the octyl-D ester derivative of D-carnosine, which had better absorption issues, failed due to dosing limitations [120]. Recent findings show supplementation of another carnosine analogue, carnosinol, alleviates carbonyl stress and diminishes some markers of diet-induced obesity in the rodent model. Carnosinol supplementation decreased HNE adduct formation in the liver and skeletal muscle, mitigated inflammation, insulin resistance and steatohepatitis [17].

Homeostasis of histidyl dipeptides within the muscle is regulated by a complex machinery of transporters (PEPT 1 and 2, TAUT), Carns1 activity, lipid peroxidation products and carnosinase 1 and 2 (CNDP), which hydrolyze carnosine to β-alanine and histidine [104,118,121]. It is not clear how these dipeptides are depleted in the muscle during metabolic syndrome. Previous studies with humans showed that the generation of lipid peroxidation products is increased and buffering capacity is diminished in the skeletal muscle of T2D subjects [39,122]. Henceforth, it is possible that histidyl dipeptides act as sacrificial peptides that are used by excessive generation of reactive aldehydes or consumed to buffer the changes in [pH]_i_. In addition, it can be speculated that the decrease in Carns1 activity might also contribute to a decrease in carnosine levels.

Despite the suggestive and circumstantial evidence showing the beneficial effects of carnosine and its analogues on metabolic syndrome, it is not clear whether the depletion of carnosine in the muscle contributes to decrease buffering capacity and is causative of insulin resistance [39]. Similarly, it is not clear whether the depletion of histidyl dipeptides affects protein folding and autophagic response within T2D muscle. Additionally, it is not clear, whether the supplementation of carnosine or its analogues to obese mice models or humans replenishes histidyl dipeptides and improves buffering capacity and autophagic responses within the muscle. Moreover, the role of enzyme Carns1, which synthesizes these dipeptides [104], has never been properly studied. Hence, a comprehensive understanding of how histidyl dipeptides regulate muscle function under physiological and pathological conditions could be elucidated by generating skeletal muscle-specific Carns-null or transgenic mice. Generation of Carns-null mice models will help to understand whether depleting histidyl dipeptides within the muscle affects skeletal muscle pH, insulin signaling and autophagy. Further, Carns overexpression in the skeletal muscle will specifically elucidate whether replenishing these dipeptides in muscle during metabolic syndrome could alleviate buffering potential and prevent protein damage and insulin resistance. Furthermore, these mice models will help understand which biochemical property, such as [pH]_i_ buffering or aldehyde quenching by these dipeptides, is essential to preserve skeletal muscle function under pathological and physiological conditions. Elucidation of these pathways will help synthesize analogues that could specifically enhance the buffering or aldehyde quenching capacity in the skeletal muscle and affect the insulin responses during metabolic syndrome.

## 6. Conclusions

Skeletal muscle comprises a large percentage of the body’s mass, handles about 70–80% of the insulin-stimulated glucose and contributes to approximately 25% of lactate formation. Derangements in glucose and fatty acid utilization within the muscle is considered to be the underlying cause of insulin resistance. In the insulin-resistant state, there is a decrease in both the uptake of glucose and key regulators of the glycolytic pathway. Studies with genetically modified mice models indicated that the overexpression or deletion of key glycolytic enzymes had either minimal effect on the obese metabolic phenotype or caused insulin resistance [32,42]. However, to fully understand, whether the defects in glycolytic pathway contributes to insulin resistance, tracer studies using ^13^C-labelled glucose are needed to understand and determine which step in the glycolytic pathway is affected and whether correction of that step could preserve insulin signaling during metabolic syndrome.

Within the skeletal muscle of T2D patients, there is increased interconversion of pyruvate to lactate [43]. Since lactate is released from the dissociation of lactic acid to lactate and H^+^ ions, increased pyruvate to lactate interconversion could drop the [pH]_i_ within the skeletal muscle of T2D patients. Additionally, a decrease in the expression of H^+^ ion transporters and histidyl dipeptide levels could have a systemic effect on the [pH]_i_ and contribute to diminish the buffering capacity within the skeletal muscle of T2D humans [39] (Figure 2). Although in vitro and in vivo studies indicate that the change in [pH]_i_ causes insulin resistance, the mechanisms by which drop in [pH]_i_ contributes to decrease insulin responses are not clearly defined. Given that skeletal muscle is the main target of insulin resistance, hence it is essential to develop a comprehensive understanding of the mechanisms, which imbalance [pH]_i_ and contribute to diminish insulin responses. An understanding of these readouts could help develop etiology-based countermeasures to alleviate insulin responses in the muscle during insulin-resistant states, such as T2D and CKD.

## Figures and Tables

**Figure 1 nutrients-12-02910-f001:**
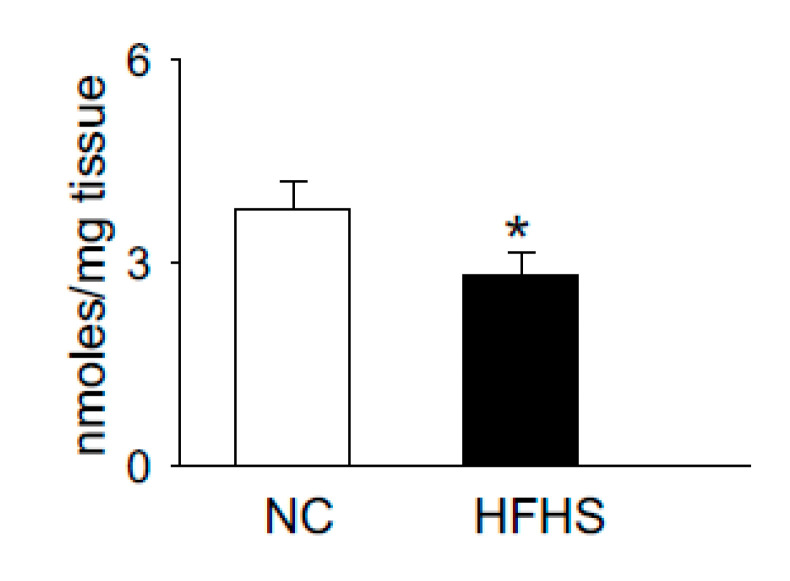
High-fat high-sucrose (HFHS) feeding to wild type (WT) mice decreases histidyl dipeptides in skeletal muscle. WT C57Bl6 mice (8 weeks old) were fed with normal chow (NC) and high-fat high-sucrose feeding for 12 weeks. Gastrocnemius muscles were isolated from the mice after NC and HFHS feeding, which were analyzed by LC–MS for anserine and carnosine levels, using tyrosine histidine as an internal standard. Data are presented as mean ± SEM, *n* = 5–6 mice in each group, * *p* < 0.05 vs NC.

**Figure 2 nutrients-12-02910-f002:**
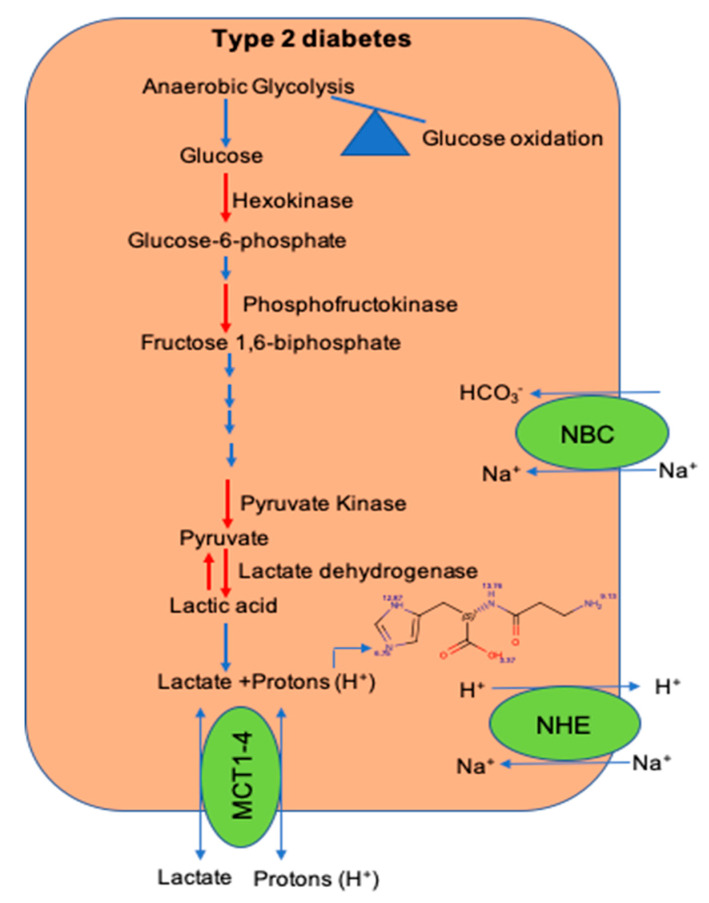
Shift in glucose utilization and imbalance of pathways that maintain intracellular pH during diet-induced metabolic syndrome. A shift in glucose utilization from glucose oxidation to anaerobic glycolysis enhances lactate production and hydrogen (H^+^) ions. Within the skeletal muscle, the activity of key glycolytic enzymes hexokinase and pyruvate kinase is decreased during metabolic syndrome. Expression of transporters; the Na-H^+^ (NHE) exchanger, which transports hydrogen (H^+^) ions during exercise and rest, respectively; and levels of histidyl dipeptides, which buffer H^+^ ions, are decreased during diet-induced metabolic syndrome.

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
