# Peer review of "Intracellular pH Regulation of Skeletal Muscle in the Milieu of Insulin Signaling"

_nutrients, 2020, doi:10.3390/nu12102910_

Round 1

Reviewer 1 Report

“Intracellular pH buffering of Skeletal Muscle in the Milieu of Insulin Signaling”

– Posa and Baba, Nutrients

Overview:

The authors present a review of literature discussing the changes in the cellular environment during the onset of insulin resistance and type 2 diabetes with an emphasis on a decrease in cellular pH and how this acidic environment may contribute to exacerbation of insulin resistance.  The authors discuss increases in lactic acid production via fermentation as well as observed decreases in buffering molecules such as histidyl dipeptides in obese and diabetic models as contributors to decreased pH in skeletal muscle.  The authors also discuss ways in which enzymes involved intramyocellular buffering are affected during metabolic disturbances, and how an overall decrease in pH during insulin resistance may contribute to the development of insulin resistance via decreased enzymatic efficiency.  The authors draw upon a wide body of literature and show good rationale for the importance of further studies examining how decreases in pH during insulin resistance occur; and how these decreases contribute to further development of metabolic disease.  This topic is of general interest and the present review is warranted.  However, the composition of the manuscript is generally confusing, and needs major reorganization and rewriting.

General comments:

The general grammar of the article makes it quite hard to follow due to missing words (e.g. line 41: “before the [pancreatic] β-cell failure”; and line 43: “insulin stimulated muscle glucose [uptake]”) and misplaced or missing commas, articles, and conjunctions (e.g. a/an/the; but/and/or).  There are also various spelling mistakes (e.g. line 110: “sequale” should be “sequelae”). Revise the article extensively for general grammar.

Although tedious, it is absolutely necessary that we continue to distinguish between proteins and mRNA/DNA by italicizing the latter. (e.g. line 99: PFK1-M should be PFKM).

Do not abbreviate words such as “approximately”.

Sectional Comments:

Abstract

Line 11: Though Covid-19 is at the center of international attention, it has little bearing upon the subject matter in this article.  Patients with obesity and type 2 diabetes have susceptibility to a variety of different negative health consequences, and the inclusion of Covid-19 specifically here does nothing valuable for the overall subject matter of the review.

Lines 22-24: “…shift from glucose oxidation to glycolysis” is confusing.  Consider revising to something like “shift from aerobic to anaerobic glycolysis” or “shift from glucose oxidation to fermentation”.

Introduction

Line 40: Metabolic inflexibility is a relatively esoteric term.  Please briefly describe what is meant by metabolic flexibility

Lines 63-65: This last sentence is confusing.  Maybe change “in the context, that” to “and discuss”

Glycolysis and Glucose Oxidation

Lines 78-80: Does this reference (#26) provide any information on Hexokinase 2 protein levels following exercise?  If so, then include this information and discuss what is meant by “posttranscriptional modification”.  Is this following translation as well?  If no information is available on the protein levels, then simply say, “postranscriptional regulation”.

Lines 100-102: Is the idea that PFKM over-expression an attempt to overcome PFK1-M allosteric inhibition evidenced in the literature, or this a personal hypothesis?” If it is the former, please expand on this idea with further evidence and cite necessary sources.  If it is the latter, please remove.  As this article reviews the contribution of altering pH to the development of insulin resistance, general interpretations of the literature should be limited to this subject. The following sentence on line 102 should also begin a new paragraph.   

Line 116: Please remove the word “surprisingly”.  Lipids are known to interfere with insulin signaling.  It is therefore not surprising that up regulating glucose oxidation and consequential downregulation of lipid metabolism would result in insulin resistance.

Line 136: New paragraph

Maintenance of Intracellular pH in the Muscle

Line 192: Provide a description of what glycolytic muscle fibers are (i.e. Type 1 versus Type 2).

Histidyl Dipeptides and Intracellular pH Regulation

This section is quite lengthy and hard to follow as a single paragraph.  The authors should reconstruct Lines 220-257 as an opening paragraph describing the importance of histidyl dipeptides as regulators of intracellular pH; then break the remaining sections into three or four paragraphs describing: 1) Their findings on diminished carnosine levels during high fat feeding in correlation with the advantages of carnosine supplementation; 2) The impact that varying levels of carnosine and other Histidyl dipeptides could have on intracellular pH, and how this might contribute to the development of insulin resistance; and 3) The importance of developing mouse models lacking enzymes within the histidyl dipeptide anabolic pathway as a means of testing alterations to intracellular pH as a contributor to the development of insulin resistance.  

Line 273: Insert parenthetical reference to Figure 1. Also, it is unclear if these data are novel or are referenced from previous work.  If the data are novel, they may need a more formal material and methods description.

Conclusions

Some remarks should be made concerning the limitations of this review.  For example, the authors should discuss the fact that it is unclear at present how an increase in H+ itself is contributing to insulin resistance in vivo.  Many of the cited studies in this review discuss the activity of enzymes measured in vitro, which may not fully recapitulate intracellular conditions.

Author Response

The authors response is attached as a Word file

Reviewer 2 Report

In this review, the authors summarize the advances in the literature in order to highlight the importance of taking an interest in the regulation of muscle pH in the context of insulin resistant states. This review is well written and clear, and addresses the problem of insulin resistance, a field that has been very studied, with an original but relevant aspect. However, some aspects deserve to be clarified and developed.

1) In my opinion, the title of the paper is not entirely representative of the content of the text. Indeed, more than insulin signaling, the paper deals with insulin resistant states and mainly type 2 diabetes. In addition, the term "intracellular pH buffering" seems a little too restrictive since it seems to exclude the part proton transport. It seems to me that intracellular pH regulation would be more appropriate.

2) In the abstract, the authors mention the implications for covid-19. While no explanatory element is given in the text on this subject, this seems superfluous. Furthermore, it does not seem necessary to justify more than necessary the importance of taking an interest in metabolic pathologies, a global health problem.

3) If the subject deals with pathologies linked to insulin resistance, other pathologies could be mentioned in the introductory part, and in particular NAFLD

4) Line 202: to my knowledge, and contrary to what is indicated, Chatel et al. demonstrated a difference in resting pH between MCT1 +/- mice and control mice. These results suggest that MCT1 has a role at rest in the entry of protons.

5) While the intramuscular pH seems to decrease in MCT1 +/- mice (Chatel et al), Lengacher et al have shown (as indicated by the authors) that these mice were protected against an HF diet. This result deserves to be discussed in more depth since it abounds in the sense of the authors, an alkaline pH  would protect against metabolic disorders, even if the underlying mechanisms are still unclear.

6) Figure 2 deserves to be much more complete, especially with many more elements of the text included. For example, the alterations of the glycolysis enzymes described in part 2 could appear to have a figure summarizing the paper as a whole.

7) Figure2: As noted in the text, MCTs can transport lactate and proton in both directions. Many studies suggest that MCT1 is rather responsible for their influxes. 

8) Part 5 seems much more developed than Part 4. However, it has been shown that protein- and HRC-related buffer capacity is not thought to play a major role in resting muscle pH regulation (juel 2008, Abe 2000, Mannion 1993), contrary to NHE1, NBC and MCT1 (Juel 2008, Aickin and Thomas 1977).

9) In this line, while the management of resting muscle pH homeostasis has been attributed mainly to the action of the sodium-proton exchanger (NHE)-1 and bicarbonate-sodium cotransporter (NBC) (Juel 2008, Aickin and Thomas 1977), the second has not been mentionned.

Author Response

The author response attached as word file

Round 2

Reviewer 1 Report

This article has greatly improved since the original submission in form and organization. In my opinion, the initial reviewer comments concerning content and literary flow have been adequately addressed.  However, the authors should work closely with the editorial staff to ensure appropriate use of English grammar prior to publication.  The most notable defeciencies are inappropriate uses of commas and articles (a, an, the), and technical deficiencies such as incomplete sentences and inappropriate punctuation (several of the words within sentences were placed within superscripts).  With sufficient modifications to the grammar of the text, this article will provide important information regarding the contribution of pH changes to the development of metabolic disease.     

Reviewer 2 Report

The authors have responded to all of my comments.

A last detail appeared to me in FIG. 2. If it is indicated lactic acid, it seems to me more relevant to indicate pyruvic acid in the previous step.

Otherwise, the article is in my opinion ready for publication.